

# Knowledge and attitudes of university staff toward organ donation: a cross-sectional study in Oman

Nasar Alwahaibi, Shahd Al Ghawi and Mohammed Al-Badi

Department of Biomedical Science, College of Medicine and Health Sciences, Sultan Qaboos University, Muscat, Oman

## ABSTRACT

**Background:** Organ donation remains low when it is not accompanied by a good knowledge and positive attitude. Most published articles have evaluated organ donation knowledge and attitudes within single categories such as healthcare workers, students, or patients. Few studies have assessed these factors across various job categories. Therefore, this study aimed to fill that gap by evaluating knowledge and attitudes about organ donation among university staff from various job categories.

**Methods:** A cross-sectional study was conducted among university staff between June 2023 and January 2024 using an online validated self-designed questionnaire. University employees were divided into academic, medical, technical and administrative staff. The survey instrument included five distinct sections: study information, informed consent, demographic data, knowledge about organ donation, and attitudes toward organ donation. Analyses included descriptive statistics, Chi-square tests, and binary logistic regression.

**Results:** The study included 385 staff. 64.4% were females, and 52.2% were in the age group between 30–41 years old. The majority of participants demonstrated good knowledge about organ donation (67.5%) and brain death (63.9%), while a significant proportion (67.5%) exhibited a negative attitude toward organ donation. Medical staff have the highest knowledge and attitude with 94.7% and 60.5%, respectively. Multivariate analysis revealed that medical and academic staff were more likely than administrative staff to have good knowledge (AOR 9.244, 95% CI [2.143–39.871]; AOR 2.300, 95% CI [1.126–4.696], respectively) and a positive attitude (AOR 3.444, 95% CI [1.633–7.262]; AOR 2.636, 95% CI [1.266–5.491], respectively), while females were 2.026 times more likely (95% CI [1.246–3.295]) to have good knowledge compared to males. The most cited organ for organ donation was kidneys (94.5%). The most common reason for supporting donating organs among university students was to save a life (67.3%) and the most common reason for refusing organs was hesitation (45.7%).

**Conclusions:** The study revealed moderate knowledge about organ donation among university staff, with medical and academic staff showing higher levels. However, overall attitudes were less positive, and there was significant reliance on the internet for information. These findings emphasize the need for targeted awareness campaigns and educational programs to improve knowledge and attitudes, promoting a cultural shift towards increased organ donation.

Corresponding author
Nasar Alwahaibi, nasar@squ.edu.om

## INTRODUCTION

Organ donation and blood donation are voluntary procedures that can help save lives and improve quality of life. However, unlike blood donation, which is accepted by almost all people and usually does not face shortages, organ donation is a real challenge in many countries globally (*Ferguson, Murray & O'Carroll, 2019*).

Despite the advanced technology in organ donation, vast experience with many transplant surgeons, the overall safety related to the transplantation, and improvement in the recipient's life, there is still a shortage of organ donations. Transplantation of organs has been shown to increase the lifespan of patients, minimize morbidity, enhance quality of life, and contribute to social and medical recovery as well as to reduce the costs associated with medical treatment (*Reese, Boudville & Garg, 2015*).

According to the International Registry in Organ Donation and Transplantation, in 2023, the organ donation rates per million population were as follows: Spain 49.38, USA 48.04, Australia 19.43, Brazil 18.65, New Zealand 12.03, Kuwait 7.67, Thailand 6.21, Saudi Arabia 4.08, Turkey 3.58, Hong Kong 3.20, Malaysia 1.25, and Oman 0.43 (*Transplantation IRiODa, 2024*). Worldwide, the willingness to donate organs varies from country to country due to factors such as culture, religion, and legal frameworks. The willingness rates for organ donation are reported as 47.5% in China, 65.7% in Saudi Arabia, 41.9% in Japan, 62% in Syria, and 67% in the United Kingdom (*Fan et al., 2022*; *Alhalafi et al., 2024*; *Cabinet Office, Government of Japan, 2017*; *Tarzi et al., 2020*; *Coad, Carter & Ling, 2013*). In contrast, 75% of people in both Iran and the USA are willing to donate their organs (*Volk et al., 2010*; *Shahbazian, Dibaei & Barfi, 2006*). In developed countries, knowledge and attitudes toward organ donation are generally positive, with high public awareness driven by educational campaigns and, in some cases, opt-out systems that presume consent for donation. However, in developing countries, awareness and attitudes vary more widely due to cultural, religious, and infrastructural factors (*Wakefield et al., 2010*). The highest organ donation rates per million population are observed in high-income countries with advanced healthcare systems, like Spain, the USA, Portugal, Belgium, and Slovenia. In contrast, developing countries, including Bangladesh, the Philippines, Vietnam, Oman, and the Dominican Republic, rank among the lowest, reflecting challenges such as limited healthcare infrastructure, cultural barriers, and lower public awareness about organ donation (*Transplantation IRiODa, 2024*). One of the efforts to increase the organ donation rates is to enhance the importance of knowledge and this might lead to a positive attitude towards organ donation. Usually, people act according to their existing knowledge or based on others' experience (*İbrahimoğlu & Urhan, 2019*; *Walker, Broderick & Sque, 2013*; *Park et al., 2010*).

Sociodemographic factors significantly influence knowledge and attitudes about organ donation. Age plays a role, with younger individuals generally having more positive attitudes compared to older generations (*Sanner, 1994*). Gender differences also exist, with women often being more supportive of organ donation than men (*Yee et al., 2021*). Higher

education levels and income are linked to better awareness and more favorable views on organ donation, as individuals with more resources tend to have greater access to information (*Wakefield et al., 2010*). Cultural, religious beliefs, and ethnicity can also affect attitudes, as certain groups may have reservations due to traditional or religious views (*Doerry et al., 2022*). In addition, geographic location matters, with urban populations typically having greater exposure to organ donation information than rural communities (*Alghanim, 2010*). Social influences, such as family and community, further shape perceptions and decisions about organ donation (*Stadlbauer et al., 2020*). Islamic teachings generally encourage saving lives, which aligns with the principle of organ donation. For instance, the Quran states, "Whoever saves a life, it is as though he had saved all mankind" (Noble Quran, Surah Al-Ma'idah, 5:32), provided it adheres to ethical and religious guidelines, such as informed consent and ensuring that it does not violate the sanctity of the donor's body.

Recently, we assessed the knowledge and attitudes regarding organ donation among 2,125 university students. The findings revealed that both the knowledge and attitudes of these students toward organ donation and transplantation were relatively low. To address this, we recommend organizing awareness campaigns and events, as well as integrating information about organ donation and transplantation into university curricula. These measures are expected to significantly improve students' knowledge and attitudes on this important subject (*Alwahaibi, Al Wahaibi & Al Abri, 2023*). Organ donation in Oman generally follows an opt-in system, meaning that organs can only be retrieved from deceased individuals who had provided prior consent by signing an organ donation form.

Measuring the knowledge and attitudes about organ donation among university staff is valuable given their influential roles within the educational community. Although university staff, both academic and non-academic, often serve as role models and educators, their educational backgrounds, exposure to information, and perspectives differ significantly from those of the general population. Consequently, the findings from this study may best reflect the perspectives of the educational community rather than the broader public. Understanding university staff's knowledge and attitudes can aid in developing targeted educational programs to address misconceptions, enhance awareness, and provide a supportive environment for organ donation within the academic setting. In addition, most published studies have evaluated organ donation knowledge and attitudes within specific groups, such as healthcare workers, students, or patients (*Kanyári et al., 2021*).

Few studies have assessed these factors across various job categories. Therefore, this study aimed to fill that gap by evaluating knowledge and attitudes about organ donation among university staff from diverse job categories.

## MATERIALS AND METHODS

### Study design

The study was conducted in all colleges, centers, hospital, deanships, libraries, and administrative units at Sultan Qaboos University. The inclusion criteria include all Omani

university staff of any age. The exclusion criteria include all staff who work in fields related to organ donation and transplantation, and whoever has either transplanted or donated an organ himself/herself or a close family member. To ensure content validity, the items were reviewed by experts. Their feedback helped refine the questions to ensure they adequately captured the constructs of knowledge and attitudes toward organ donation. The questionnaire was conducted online *via* Google Forms. The survey was distributed through a combination of direct emails, social media, and self-administered to ensure a wide reach. Emails were sent to participants identified through an institutional database and were not cold emails. Social media distribution targeted closed and relevant professional groups to engage the intended audience effectively. Self-administered forms were made available at specific locations within the institution to ensure broader participation. To minimize potential biases associated with these methods, we ensured response anonymity, used a standardized introduction explaining the study purpose, and sent consistent follow-up reminders. The questionnaire contains some self-developed questions, and others were obtained from literature reviews (*Alwahaibi, Al Wahaibi & Al Abri, 2023*; *Asimakopoulou et al., 2021*; *Mohsin et al., 2010*). Additionally, the questionnaire was available in both English and Arabic versions. The Arabic and English versions of the questionnaire were carefully proofread by native speakers of each language to ensure the accuracy and clarity of the translations.

## Study setting

Sultan Qaboos University, which was established in 1986, is a public university located in al-Seeb, Oman. By the end of 2022, it has 5,570 staff including 3,922 Omani (2,036 males and 1,886 females) and 1,648 non-Omani. It has five scientific and four humanities colleges, 13 services centers, 14 research centers, four deanships, and one hospital. The total number of students is 18,858 including 17,113 undergraduate and 1,745 postgraduate students.

## Sample size calculation

The sample size was calculated by using the formula $n = NZ2p\ (1 - p)/\{d^2(N - 1) + Z^2p\ (1 - p)\}$, where n = sample size, N = total number of SQU staff = 3,922, Z = standard normal deviate = 1.96 with a confidence level of 95%, d = permissible error on each side of 2%, and p = prevalence from a previous study with a 0.34 (*Alwahaibi, Al Wahaibi & Al Abri, 2023*). Moreover, 20% was added to the sample size to avoid any incorrectly filled-out questionnaires. As a result, the sample size consisted of 385 participants. For reliability testing, a pilot study was conducted among 20 university staff from different colleges, centers, hospital, deanships, libraries, and administrative units who fulfilled the research criteria. Those who participated in the pilot study were excluded from the study. This allowed us to refine the wording and structure of the questionnaire based on their feedback. The Cronbach's alpha for the reliability of the questionnaire for knowledge and attitude was 0.706 and 0.609, respectively. While the attitude scale's Cronbach's alpha is slightly below the commonly accepted threshold of 0.7, it is still within an acceptable range for exploratory studies.

## Data collection

The questionnaire composed of five sections. The first section provided essential information about the study, including its purpose, objectives, procedures, ethical approval, and the importance of participation. The second section included the informed consent form, which participants were required to review and accept before proceeding. The third section collected sociodemographic data such as sex, age, marital status, job title, years of work experience, and primary sources of information about organ donation. The fourth section assessed participants' knowledge of organ donation, covering concepts such as the definition and purpose of organ donation, the possibility of donating whole or partial organs, associated benefits and risks, and awareness of national programs or registries. The final section explored attitudes toward organ donation, including beliefs about its role in saving lives, willingness to register as a donor if a national registry were available, support for promoting organ donation among family members, and perceived barriers to donation. To enhance response accuracy and reduce bias, the questionnaire included a mix of positively and negatively worded items. In this study, a random sampling method was employed. To enhance the response accuracy and ensure data quality, we implemented several strategies. First, the questionnaire was validated to ensure clarity and relevance, minimizing response bias. Detailed instructions were provided to guide respondents in completing the survey accurately, and confidentiality was assured to encourage honest and accurate answers. Finally, after collecting the surveys, data cleaning was conducted to identify and remove incomplete entries, ensuring the integrity of the dataset.

## Scores of knowledge and attitude

Fifteen questions were employed to evaluate participants' knowledge of organ donation, with response options including "yes," "no," and "I do not know." We chose the "yes," "no," and "I don't know" options to simplify data collection and ensure participants could respond without confusion, especially given the diverse educational backgrounds of our respondents. This format allowed us to capture clear and straightforward responses. Scores ranged from zero to 15. Participants scoring <60% (0–8 out of 15) were classified as having a poor knowledge, while those scoring ≥60% (9–15 out of 15) were categorized as having a good knowledge (*Asimakopoulou et al., 2021*). Attitudes were assessed through ten questions with response options "yes," "no," or "I don't know," along with two multiple-choice questions. Similar scoring criteria as for knowledge were used for attitudes, where achieving 60% or more indicated a positive attitude, while less than 60% indicated a negative attitude (*Asimakopoulou et al., 2021*). The two multiple-choice questions were designed to ascertain factors influencing participants' willingness or refusal to donate organs. 'I don't know' answers were combined with the 'no' in the association tables for both knowledge and attitudes. This approach was adopted to simplify the interpretation of the data, as 'I don't know' responses are indicative of a lack of knowledge or certainty, aligning more closely with a 'no' response in the context of knowledge and attitude assessment.

## Data analysis

The data were analyzed using Statistical Package for Social Science (SPSS) version 25 software (IBM Corp., Armonk, NY, USA). Frequencies and percentages were used to represent categorical data such as gender, age, marital status, academic degree, and working experience. Continuous variables were expressed using the mean and standard deviation. Associations between sociodemographic factors and levels of knowledge and attitudes regarding organ donation were evaluated using the Chi-square test. Binary logistic regression was used for multivariate adjusted analysis. Factors with $p$-value of $< 0.25$ in the crude analysis were included in the multivariate analysis. The $p$-value was considered significant if it was less than 0.05 only.

## Ethical consideration

The study was conducted in accordance with the guidelines of the Declaration of Helsinki and approved by the Medical Research Ethics Committee (MREC), College of Medicine and Health Sciences, Sultan Qaboos University, Oman, with an ethical approval number MREC #2920. This cross-sectional observational study was conducted between June 2023 and January 2024. Written informed consent was obtained from all participants. Prior to the study, the procedure was thoroughly explained to cover the study's objectives, confidentiality, and anonymity of their participation. Additionally, participants' right to refuse involvement in the study was respected.

## RESULTS

A total of 415 individuals participated in the study, but only 385 met the inclusion criteria, with 30 participants excluded for not fulfilling the criteria. The response rate was 92.77%. Among the 385 participants, 190 (49.35%) were administrative staff, 38 (9.87%) were medical staff, 102 (26.49%) were technical staff, and 55 (14.29%) were academic staff. Females made up 64.4% of the participants, and the majority were between 31 and 41 years old. In addition, 77.4% were married, 63.4% had undergraduate degrees or lower qualifications, and 50.1% had less than 11 years of work experience (Table 1).

The majority of university staff (67.53%) demonstrated good knowledge about organ donation and brain death, while a smaller portion (32.47%) had poor knowledge. Furthermore, most were aware that Islam permits organ donation (71.2%), nearly all had heard about it (98.4%), and a high percentage (90.1%) understood that organ donation can save lives. The most cited organs that can be donated were kidneys, portion of the liver, and bone marrow with 94.5%, 78.2%, and 62.9%, respectively. University staff showed 63.9% good knowledge about overall brain death, and 92.7% heard about brain death. 90.1% knew that rejection of the transplanted organ is possible and 57.4% knew where to register to donate organs after death (Table 2).

Females were significantly higher in good knowledge compared to males (75% $vs.$ 54%; $\chi^2(1) = 16.780$, $p = 0.001$). A significant association was also observed with job title ($\chi^2(3) = 17.503$, $p = 0.001$), where medical staff had the highest proportion of good knowledge (94.7%), followed by academic staff (74.5%), technical staff (64.7%), and administrative staff (61.6%). Years of work experience were significantly associated with

**Table 1 The sociodemographic characteristics about organ donation among university staff at Sultan Qaboos University (N = 385), Oman.**

|  | Characteristics | Number | Percent |
|---|---|---|---|
| Gender | Males | 137 | 35.6 |
|  | Females | 248 | 64.4 |
| Age group | 18–29 | 54 | 14.0 |
|  | 30–41 | 201 | 52.2 |
|  | 42–53 | 113 | 29.4 |
|  | 54–65 | 17 | 4.4 |
| Marital status | Single | 75 | 19.5 |
|  | Married | 298 | 77.4 |
|  | Divorced | 12 | 3.1 |
| Education | Undergraduate or less |  |  |
|  | Primary school | 3 | 0.8 |
|  | Secondary school | 3 | 0.8 |
|  | Diploma degree | 57 | 14.8 |
|  | Bachelor's degree/Doctor of Medicine (MD) | 181 | 47.0 |
|  | Master's degree | 82 | 21.3 |
|  | PhD's degree | 59 | 15.3 |
| Administrative staff | Head | 30 | 7.8 |
|  | Director | 8 | 2.1 |
|  | Deputy director | 4 | 1.0 |
|  | Administrator | 109 | 28.3 |
|  | Coordinator | 39 | 10.1 |
| Medical staff | MD | 19 | 4.9 |
|  | Nurse | 10 | 2.6 |
|  | Biomedical scientist | 5 | 1.3 |
|  | Medical orderly | 2 | 0.5 |
|  | Pharmacist | 2 | 0.5 |
| Technical staff | Chief technician | 11 | 2.9 |
|  | Senior technician | 20 | 5.2 |
|  | Technician | 39 | 10.1 |
|  | Engineer | 9 | 2.3 |
|  | Researcher | 8 | 2.1 |
|  | Chief librarian | 1 | 0.3 |
|  | Data entry | 2 | 0.5 |
|  | Programmer | 2 | 0.5 |
|  | Graphic designer | 4 | 1.0 |
|  | Accountant | 3 | 0.8 |
|  | Driver | 1 | 0.3 |
|  | University's security | 2 | 0.5 |
| Academic staff | Professor | 3 | 0.8 |
|  | Assistant professor | 25 | 6.5 |

(Continued)

| | Characteristics | Number | Percent |
|---|---|---|---|
| | Associate professor | 14 | 3.6 |
| | Lecturer | 11 | 2.9 |
| | Instructor | 2 | 0.5 |
| Number of working years | 1–11 | 193 | 50.1 |
| | 12–23 | 139 | 36.1 |
| | 24–35 | 53 | 13.8 |

**Table 2 University staff' responses to knowledge level towards organ donation among university staff (N = 385) at Sultan Qaboos University, Oman.**

| | Questions | Variables | Responses |
|---|---|---|---|
| 1 | Have you heard about organ donation? | Yes | 98.4% (379) |
| | | No | 1.6% (6) |
| 2 | Did the "Islamic Fatwa" allow organ donation? | Yes | 71.2% (274) |
| | | No | 2.6% (10) |
| | | I do not know | 26.2% (101) |
| 3 | Would donated organs save other people's lives? | Yes | 90.1% (347) |
| | | No | 1.3% (5) |
| | | I do not know | 8.6% (33) |
| 4 | Which of the following organs could be donated by live healthy donors: | A kidney | 94.5% (364) |
| | | A lung or part of it (lobe) | 30.4% (117) |
| | | Portion of the liver | 78.2% (301) |
| | | Portion of pancreas | 22.3% (86) |
| | | Portion of intestine | 18.4% (71) |
| | | Bone marrow | 62.9% (242) |
| | | Cornea | 25.7% (99) |
| 5 | Have you heard about brain death? | Yes | 92.7% (357) |
| | | No | 7.3% (28) |
| 6 | In brain death, do all the brainstem reflexes stop? | Yes | 24.4% (94) |
| | | No | 32.5% (125) |
| | | I do not know | 43.1% (166) |
| 7 | In brain death, the heart can still beat | Yes | 75.8% (292) |
| | | No | 5.2% (20) |
| | | I do not know | 19.0%% (73) |
| 8 | Is brain death irreversible? | Yes | 36.9% (142) |
| | | No | 13.0% (50) |
| | | I do not know | 50.1% (193) |
| 9 | Can organs be transplanted from patients with brain death? | Yes | 55.3% (213) |
| | | No | 4.7% (18) |
| | | I do not know | 40.0% (154) |
| 10 | Do you know any information about commercial transplanting? | Yes | 61.8% (238) |
| | | No | 38.2% (147) |

| Questions | Variables | Responses |
|---|---|---|
| 11 Are there any laws regarding organ donation and transplantation in Oman? | Yes | 56.6% (218) |
| | No | 1.8% (7) |
| | I do not know | 41.6% (160) |
| 12 The donor and recipient's blood groups must be identical. | Yes | 45.5% (175) |
| | No | 15.3% (59) |
| | I do not know | 39.2% (151) |
| 13 Kidney is the most commonly transplanted organ. | Yes | 69.4% (267) |
| | No | 0.8% (3) |
| | I do not know | 29.9% (115) |
| 14 Rejection of an organ after transplantation is possible. | Yes | 90.1% (347) |
| | No | 0.8% (3) |
| | I do not know | 9.1% (35) |
| 15 Do you know that you can register to donate your organ after death through AL-Shifa app? | Yes | 57.4% (211) |
| | No | 42.6% (174) |

knowledge ($\chi^2(2) = 11.644$, $p = 0.002$), with better knowledge observed among staff with fewer than 23 years of experience. In contrast, age group ($\chi^2(3) = 1.985$, $p = 0.576$), marital status ($\chi^2(2) = 0.803$, $p = 0.669$), and academic qualification ($\chi^2(1) = 0.265$, $p = 0.607$) were not significantly associated with knowledge levels (Table S1).

However, the multivariate analysis showed that gender and job title were independent factors for knowledge level. In this regard, females were 2.026 (95% CI [1.246–3.295]) times more likely to have good knowledge compared to males, with $p$-value of 0.004. In addition, medical staff and academic staff were also more likely to have good knowledge compared to administrative staff, with AOR of 9.244 (95% CI [2.143–39.871]), and 2.300 (95% CI [1.126–4.696]), respectively. The logistic regression model was statistically significant ($\chi^2(6) = 40.6$, $p < 0.001$), indicating that the included predictors significantly improved the model's ability to classify knowledge levels. The model explained approximately 14.0% of the variance in knowledge ($R^2 = 0.140$). Full results are presented in Table 3.

In terms of attitudes, a positive attitude towards organ donation was observed in 36.7% of staff, while the majority (63.3%) held a negative attitude. In addition, 35.3% indicated willingness to donate their kidneys or other organs after death, and 38.2% would consider registering as donors. However, only a small percentage (3.1%) had already registered, and 71.4% expressed refusal to sell their organs for money (Table 4).

Medical staff had the highest positive attitude toward organ donation (60.5%), followed by academic staff (52.7%). Among marital status groups, divorced participants showed the most favorable attitude (58.3%). Statistically significant associations were found between attitude and both marital status ($\chi^2 = 7.095$ (2), $p = 0.029$) and job title ($\chi^2 = 18.386$ (3) $p = 0.001$). In contrast, age group ($\chi^2 = 7.228$ (3), $p = 0.065$), sex ($\chi^2 = 2.101$ (1), $p = 0.147$), academic degree ($\chi^2 = 1.657$ (1), $p = 0.198$), and years of work experience ($\chi^2 = 3.836$ (2),

**Table 3 Multivariate results for factors affecting knowledge level towards organ donation among university staff (N = 385) at Sultan Qaboos University, Oman.**

| Factors | Categories | AOR | 95% CI lower limit | 95% CI upper limit | p-value |
|---|---|---|---|---|---|
| Sex | Males | Reference | | | 0.004 |
| | Females | 2.026 | 1.246 | 3.295 | |
| Job title | Administrative staff | Reference | | | Overall p 0.004 |
| | Medical staff | 9.244 | 2.143 | 39.871 | 0.003 |
| | Technical staff | 1.239 | 0.738 | 2.078 | 0.417 |
| | Academic staff | 2.300 | 1.126 | 4.696 | 0.022 |
| Number of working years | 1–11 | Reference | | | Overall p 0.112 |
| | 12–23 | 0.961 | 0.580 | 1.594 | 0.879 |
| | 24–35 | 0.487 | 0.242 | 0.982 | 0.044 |

Note:
AOR, adjusted odds ratio; CI, confidence interval.

**Table 4 University staff' responses to attitude level towards organ donation among university staff (N = 385) at Sultan Qaboos University, Oman.**

| | | Yes | | No | | I do not know | |
|---|---|---|---|---|---|---|---|
| | | Number | Percent | Number | Percent | Number | Percent |
| 1 | Would you donate your kidneys or other organs after death? | 136 | 35.3 | 63 | 16.4 | 186 | 48.3 |
| 2 | If a member of your family develops kidney failure, will you donate your kidney to him or her? | 269 | 69.9 | 21 | 5.5 | 95 | 24.7 |
| 3 | If you or a member of your family develops renal failure, would you accept a kidney from a deceased person? | 258 | 67.0 | 30 | 7.8 | 97 | 25.2 |
| 4 | Would you like to register to donate your organs after death? | 147 | 38.2 | 74 | 19.2 | 164 | 42.6 |
| 5 | Do you think that organ donation is ethically and legally true? | 274 | 71.2 | 15 | 3.9 | 96 | 24.9 |
| 6 | Do you agree with organ donation from non-relatives? | 237 | 61.6 | 46 | 11.9 | 102 | 26.5 |
| 7 | Do you agree with organ donation if you have financial interest? | 26 | 6.8 | 275 | 71.4 | 84 | 21.8 |
| 8 | Do you think someday you may need organ transplant? | 54 | 14.0 | 59 | 15.3 | 272 | 70.6 |
| 9 | Have you registered for organ donation? | 12 | 3.1 | 356 | 92.5 | 17 | 4.4 |
| 10 | Have you donated or received an organ before? | 0 | 0.0 | 368 | 95.6 | 17 | 4.4 |

$p = 0.147$) were not significantly associated with attitudes toward organ donation (Table S2). For the multivariate analysis for factors affecting the attitude, the job title was the only independent factor. In this regard, medical staff and academic staff were 3.444 (95% CI [1.633–7.262]) and 2.636 (95% CI [1.266–5.491]) times more likely to have good attitude compared to administrative staff, with $p$-values of 0.001 and 0.010 respectively. Marital status was not an independent factor for attitude level. The binary logistic regression model was statistically significant, $\chi^2 = 37.364$ (12), $p = 0.001$, indicating that the model significantly predicted the outcome. The $R^2$ was 0.126 (12.6%), suggesting that the model explained approximately 16.8% of the variance in attitude toward organ donation (Table 5).

The most common reason for supporting donating organs among university staff was to save a life, followed by compliance with the Islamic religion, and to become a donor for

**Table 5 Multivariate results for factors affecting attitude level towards organ donation among university staff (N = 385) at Sultan Qaboos University, Oman.**

| Factors | Categories | AOR | 95% CI lower limit | 95% CI upper limit | p-value |
|---|---|---|---|---|---|
| Sex | Males | Reference | | | 0.638 |
| | Females | 1.127 | 0.685 | 1.856 | |
| Age groups | 18–29 | Reference | | | Overall p 0.054 |
| | 30–41 | 0.377 | 0.184 | 0.771 | 0.008 |
| | 42–53 | 0.406 | 0.160 | 1.029 | 0.057 |
| | 54–65 | 0.638 | 0.133 | 3.051 | 0.573 |
| Marital status | Single | Reference | | | Overall p 0.157 |
| | Married | 0.660 | 0.359 | 1.214 | 0.182 |
| | Divorced | 1.697 | 0.440 | 6.542 | 0.443 |
| Academic degree | Undergraduate or less | Reference | | | 0.410 |
| | Postgraduate | 1.255 | 0.731 | 2.153 | |
| Job title | Administrative staff | Reference | | | Overall p 0.002 |
| | Medical staff | 3.444 | 1.633 | 7.262 | 0.001 |
| | Technical staff | 1.268 | 0.746 | 2.155 | 0.381 |
| | Academic staff | 2.636 | 1.266 | 5.491 | 0.010 |
| Number of working years | 1–11 | Reference | | | Overall p 0.124 |
| | 12–23 | 1.409 | 0.783 | 2.537 | 0.253 |
| | 24–35 | 0.623 | 0.220 | 1.764 | 0.373 |

**Note:**
  AOR, adjusted odds ratio; CI, confidence interval

someone dear to me, with 67.3, 46.8, and 23.6%, respectively (Table S3). Significant associations were found between age group and supporting organ donation based on Islamic beliefs ($p = 0.044$, $\chi^2 = 8.111$ (3)). The majority of respondents in the 30–41 age group (50.6%) and 42–53 group (33.3%) endorsed this reason, indicating a stronger influence of religious justification among middle-aged staff. Job title was significantly associated with support for organ donation as a life-saving act ($p = 0.046$, $\chi^2 = 7.997$ (3)). Administrative staff accounted for the largest proportion (44.8%) of those endorsing this reason, suggesting occupational role may influence attitudes toward donation promotion. No other sociodemographic variables showed statistically significant associations with these reasons for supporting organ donation (Table S4).

The most common reason for refusing donating organs among university staff was hesitation with 45.7%, and 40.8% have no objection for organ donation (Table S5). Fear was a significantly more common reason for refusing organ donation among younger staff aged 30–41 years (54.9%) compared to other age groups ($\chi^2 = 12.903$ (3), $p = 0.005$). Staff with an undergraduate or lower academic qualification were significantly more likely to cite fear as a reason (74.5%) than those with postgraduate degrees ($\chi^2 = 7.410$ (1), $p = 0.006$). In addition, fear was significantly higher among administrative staff (58.8%) compared to other job titles ($\chi^2 = 11.851$ (3), $p = 0.008$), and among those with fewer years of work experience ($\chi^2 = 7.539$ (2), $p = 0.023$). In contrast, refusing organ donation due to religious beliefs ("against Islamic religion") did not show any statistically significant association with sociodemographic characteristics (Table S6).

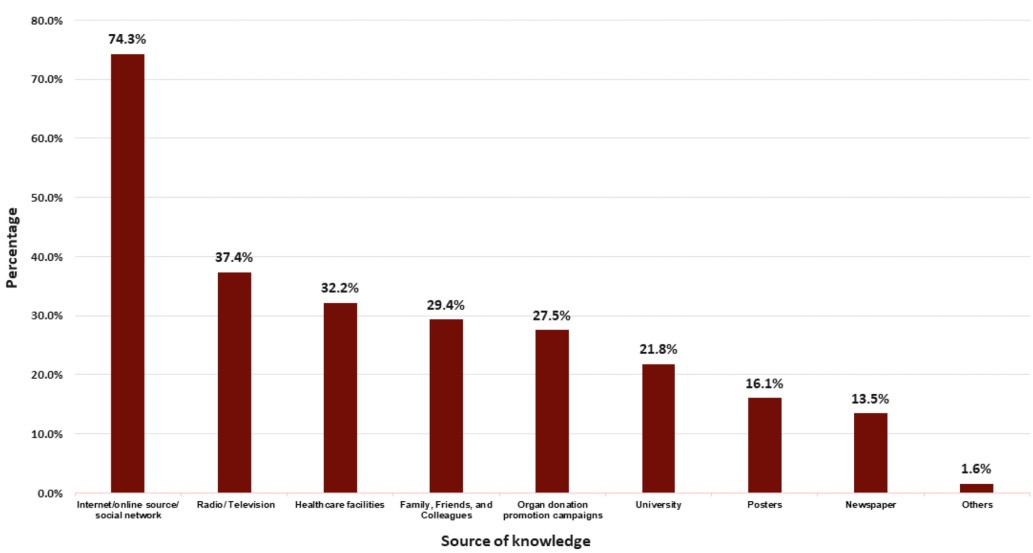

**Figure 1 Sources of knowledge about organ donation among university staff at Sultan Qaboos University, Oman.**

In the questionnaire, participants were asked to provide reasons for either supporting or refusing organ donation. An open-ended 'Others' option was included to allow participants to specify any additional reasons not covered in the predefined choices. However, many participants did not provide responses in the text box. As a result, the 'Others' category was left empty for those who chose it, and no further details were recorded. The most common sources of information about organ donation among university staff were internet, online sources and social networks represented by 74.3%, followed by radio and TV 37.4%, and healthcare facilities 32.2%. There was a limited amount of knowledge about organ donation available through posters and newspapers, 16.1, and 13.5%, respectively (Fig 1). A statistically significant relationship was found between Internet/online sources/social networks and knowledge levels ($\chi^2$ = 12.783(1), $p < 0.001$), with 72.7% of those exposed demonstrating good knowledge. In contrast, other sources such as university, healthcare facilities, newspapers, and radio/TV showed no significant associations with knowledge. For attitudes, statistically significant associations were observed with posters ($\chi^2$ = 6.855(1), $p = 0.009$) and organ donation promotion campaigns ($\chi^2$ = 6.204(1), $p = 0.013$), suggesting that these sources may positively influence participants' attitudes. No significant associations were noted between attitudes and other sources, including internet, university, and healthcare facilities (Table S7).

## DISCUSSION

This study involves various job categories including academicians, physicians, nurses, biomedical scientists, engineers, technicians, administrators, medical orderlies, drivers, coordinators, and others. University staff usually come from different regions, have different knowledge and qualifications, and work in different places with various rules and positions. The aim of this study was to evaluate the knowledge and attitude about organ donation among university staff.

The findings of this study showed that knowledge about organ donation among university staff was 67.53%. Other studies reported similar or higher scores (*Ibrahim & Randhawa, 2017*; *Yadav et al., 2020*). Good knowledge was more prevalent among the medical (94.7%) and academic staff (74.5%). Medical staff are expected to have basic knowledge of organ donation due to their extensive educational programs and training. Their knowledge is important not only to guide, but also to encourage donors for organ donation. Healthcare workers in Turkey and Nigeria demonstrated good knowledge for organ donation with 87.3% and 63.9%, respectively (*Okpere & Anochie, 2014*; *Akgun et al., 2003*). In contrast, other studies reported poor knowledge among healthcare workers regarding organ donation (*Alsultan, 2012*; *Evanisko et al., 1998*; *Molzahn, 1997*). These insights highlight the importance of targeted educational programs to improve organ donation awareness. The present study found an association between sex and knowledge. Females (75%) are more knowledgeable than males (54%). Other study reported similar finding (*Pouraghaei et al., 2015*). However, other studies did not find an association between sex and knowledge (*Oktem et al., 2020*; *Yazar & Açıkgöz, 2016*).

The present study also showed higher participation among female staff (64.4%) compared to male staff (35.6%). This is slightly different from the university's overall staff composition, where the gender ratio is 51.9% men and 48.1% women. This discrepancy could be due to the recruitment method and varying response rates among different staff groups. Similar studies have also reported higher female participation in organ donation surveys (*Soqia et al., 2023*; *Fan et al., 2022*; *Somaili et al., 2022*). Female participation in organ donation knowledge and attitudes surveys is often higher due to several factors. Women usually have greater health awareness and engagement in health-related activities, leading to increased involvement in such surveys. Women often show greater altruism and empathy, which motivates them to participate in socially beneficial causes like organ donation (*Hallyburton & Evarts, 2014*). Their active involvement in community and social networks increases their exposure to information and campaigns about organ donation (*Rowley, Johnson & Sbaffi, 2015*). Moreover, women generally possess better communication skills and are more comfortable expressing their opinions, which further enhances their survey participation rates (*Stern, Cotten & Drentea, 2011*). The inconsistent association between sex and organ donation knowledge across studies may reflect variations in sample composition, cultural norms, response biases, and measurement tools. In some contexts, females demonstrate greater health awareness and survey participation, while in others, differing educational or professional exposures may favor male respondents. Thus, sex-based differences should be interpreted cautiously and within the broader context of demographic and methodological variability.

The findings of this study reveal that the attitude of university staff toward organ donation was only 35.6%. This finding is lower than other reported studies. For example, a study of academic, administrative staff, and their relatives from four different universities in Turkey (Ankara, Eskisehir, Istanbul and Adana), found that 69.1% of the participants were willing for being organ donors (*Oktem et al., 2020*). In a Malaysia, 383 patients from an outpatient clinic, showed that 195 (50.9%) respondents had a positive attitude toward organ donation (*Lim et al., 2020*). In Syria, 58% of 303 survey participants expressed a

willingness to donate their organs (*Tarzi et al., 2020*). Another recent study in Syria reported almost similar finding (62.8%) (*Soqia et al., 2023*). The attitudes toward organ donation among 1,052 non-medical staff members surveyed in different hospitals in Cuba, Mexico, Costa Rica, and Spain were 98%, 80%, 66%, and 52%, respectively (*Ríos et al., 2013*). Other studies in Jordan, South Korea, Morocco and Saudi Arabia reported comparable results with 72%, 60.9%, 57.6%, and 42%, respectively (*Al-Qerem, Carter & Ling, 2022*; *Lee et al., 2017*; *El Hangouche et al., 2018*; *Alam, 2007*). In addition, a recent systematic review on organ donation by *Al-Abdulghani et al. (2024)* found that a favorable attitude toward organ donation was positively correlated with individuals' willingness to donate. However, the medical staff in the current study showed 60.5% willingness to donate their organs. This finding is higher than other two studies where the medical staff in Turkey and Nigeria showed only 44.2% and 50.7% willingness to donate organs, respectively (*Okpere & Anochie, 2014*; *Akgun et al., 2003*). This discrepancy suggests that while medical professionals generally show more support for organ donation, there is still a need to enhance awareness and positive attitudes toward organ donation among university staff and the general population. We found no association between sex and attitude. This finding is in line with other study (*Yadav et al., 2020*). However, other studies have reported that females had a positive attitude (*Mekahli et al., 2009*; *Burra et al., 2005*).

University staff in this study identified the kidney as the most commonly donated organ for living donation (94.5%). In agreement with this finding, 82% of healthcare professionals in the Saolta University Health Care Group (University Hospital Galway, Sligo University Hospital, Mayo University Hospital, Roscommon Hospital, Letterkenny University Hospital and Portiuncula University Hospital), Ireland, stated that the kidney was the most common organ for living donation in their survey (*Umana et al., 2018*). Other studies reported similar results (*Alghamdi et al., 2023*; *Altraif et al., 2020*; *Yadav et al., 2020*). This widespread agreement highlights the strong recognition of kidney donation as a prevalent practice in living organ donation, reflecting a common understanding across different regions and healthcare settings.

The present study showed that 45.7% of university staff refused to donate organ donations because they are not yet decided about organ donations, while 40.8% had no objection with organ donation, and 26.2% were afraid to donate organs. The findings indicate that fear remains a significant barrier to organ donation acceptance among university staff, particularly among younger individuals, those with lower academic qualifications, administrative staff, and those with fewer years of work experience. In line with this study, 40.3% of the general population in Limassol, Cyprus, reported that fear was the main reason for participants not intending to be organ donors (*Asimakopoulou et al., 2021*). Another study in Saudi Arabia that was conducted to assess the public perception on organ donation and transplantation, showed that 15.2% expressed fear of the operation (*Mohamed & Guella, 2013*). In Turkey, the most important reason for not wanting to donate organs was that participants (40.9%) do not want their body to lose their integrity (*Şenyuva, 2022*). While in Syria, the most common reason to refuse donation was the refusal to disfigure a dead body by removing an organ (41%) (*Soqia et al., 2023*). Overall, these insights highlight that while fear and concerns about body integrity are common

barriers, a substantial number of individuals remain open to organ donation or undecided, indicating potential areas for targeted education and reassurance. Saving a life is the main reason for organ donation among the university staff (67.3%). Similar findings were reported (*Akbulut et al., 2022*; *Şenyuva, 2022*; *Asimakopoulou et al., 2021*; *Yazar & Açıkgöz, 2016*; *Tsavdaroglou et al., 2013*). Also, in our previous study, when we asked the university students what supports them to donate organs, 76.8% say that they want to save a life (*Alwahaibi, Al Wahaibi & Al Abri, 2023*). These consistent results show a strong and widespread commitment to organ donation, driven by the goal of saving lives. This reflects ongoing support for organ donation across different studies. In addition, the current findings highlight key sociodemographic factors influencing donation motivations. A significant association between age and citing Islamic beliefs suggests that middle-aged individuals may be more guided by religious considerations when forming attitudes toward donation. Furthermore, the significant association between job title and the belief that donation saves lives suggests varying levels of awareness or engagement across occupational roles. Administrative staff, in particular, may be more receptive due to greater exposure to institutional health messaging. These findings underscore the need for targeted educational interventions tailored to age groups and job categories to enhance awareness and support for organ donation.

The present study found that 63.9% of the university staff had good knowledge about brain death. This finding is higher than findings from studies in Malaysia and Syria, where only 32.6 and 40% of participants had good knowledge of brain death, respectively (*Lim et al., 2020*; *Tarzi et al., 2020*). Donors are less likely to donate organs when they have no knowledge about brain death (*Abbasi et al., 2020*). There is no doubt that brain-dead people are the most important source of organ donations. In fact, studies have found that attitudes toward donation are significantly affected by accepting brain death as a valid term of death (*Hu & Huang, 2015*; *Cohen et al., 2008*; *Rios et al., 2005*). University staff can influence their friends, families, and societies in all aspects of organ donation. Academic staff can encourage the under and postgraduate students, medical staff encourage patients (who can donate) and their relatives, technical and administrative staff can encourage their colleagues in different areas and workplaces. Thus, the effectiveness of organ donation awareness can reach many people across the country.

The findings of this study showed that university staff relied mainly on the internet, online sources and social networks as a source of information about organ donation (74.3%). This aligns with our previous study among university students, where 84.13% reported similar sources of information (*Alwahaibi, Al Wahaibi & Al Abri, 2023*). Comparable patterns were observed in Turkey and Brazil, where internet and media platforms were among the most cited sources (*Videira et al., 2024*; *Akbulut et al., 2022*). Moreover, statistical analysis confirmed a significant association between online information sources and better knowledge ($p = 0.001$, $\chi^2 = 12.783$ (1)), highlighting their impact in academic settings. In addition, posters and organ donation campaigns were significantly associated with positive attitudes ($p = 0.009$, $\chi^2 = 6.855$ (1); $p = 0.013$, $\chi^2 = 6.204$ (1)), suggesting that visual and targeted outreach efforts effectively promote favorable perceptions. In contrast, traditional sources like television, newspapers, and

radio showed limited influence, indicating a need to prioritize digital and campaign-based strategies to enhance awareness and attitudes toward organ donation.

The findings highlight the need for targeted strategies to improve organ donation awareness, especially among administrative staff and male employees. Educational sessions, culturally sensitive materials, and integration of donation topics into orientation and staff development programs can enhance knowledge and attitudes. Collaborating with religious and healthcare leaders may address ethical concerns. Institutions should embed organ donation awareness into broader health initiatives, supported by leadership, through activities like awareness weeks or inclusion in employee wellness events.

A key strength of this study is its inclusion of a diverse range of job categories, including academicians, physicians, nurses, biomedical scientists, engineers, technicians, administrators, medical orderlies, drivers, coordinators, and others. This broad representation enhances the study's ability to capture a wide variety of perspectives and experiences related to organ donation across different professional sectors.

The study has some limitations. First, this study was conducted only among one single university. Even that this university is the only national government university in Oman. Thus, to generalize the findings, the study should include staff from other universities. Second, the distribution of participants between different jobs are not equal as some jobs have more responses than others. Thirdly, during survey assembling, there is a possibility of interaction between staff. Fourthly, the exclusion of staff with direct organ donation or transplantation experience, which may omit valuable insights, future research should consider including these groups for comparative analysis. Fifthly, the use of yes/no responses to measure attitudes, which may lack distinction, future studies could improve validity by using a semantic or Likert scale. Sixth, the questionnaire was distributed online, more accurate results might have been obtained with face-to-face surveys. Finally, as with any self-reported data, particularly on sensitive topics like organ donation, there is a risk of response and social desirability bias. Participants may have provided answers they perceived as socially acceptable rather than their true beliefs or behaviors, which could affect the interpretation of knowledge and attitude levels.

## CONCLUSION

The present study has provided valuable insights into the knowledge and attitudes toward organ donation among university staff, including a diverse range of job categories. The findings indicated that while the overall knowledge about organ donation was moderate, there were notable differences across job roles, with medical and academic staff demonstrating higher levels of knowledge. Conversely, administrative staff and male participants were identified as groups with relatively lower knowledge levels, highlighting the need for targeted interventions. The overall attitude towards organ donation was less positive. The study also identified significant reliance on internet and online sources for information about organ donation. It also highlighted the importance of addressing fears about organ donation. These findings underscore the need for comprehensive educational programs within university settings that go beyond simple awareness. Such programs

should focus on deepening knowledge, addressing specific fears, and fostering positive attitudes toward organ donation, ultimately promoting a cultural shift in perceptions and willingness to donate that could contribute to increased organ donation rates.

## ACKNOWLEDGEMENTS

The authors would like to thank all participants in this study. The authors acknowledge the use of AI (GPT 3.5) to support language editing and formatting.

### Funding

Financial support to conduct the study was obtained from the Deanship of Research at Sultan Qaboos University in Oman, with the number RF/MED/BIOM/23/01. The funders had no role in study design, data collection and analysis, decision to publish, or preparation of the manuscript.

### Grant Disclosures

The following grant information was disclosed by the authors:
Deanship of Research at Sultan Qaboos University in Oman: RF/MED/BIOM/23/01.

### Competing Interests

The authors declare that they have no competing interests.

### Author Contributions

- Nasar Alwahaibi conceived and designed the experiments, performed the experiments, analyzed the data, prepared figures and/or tables, authored or reviewed drafts of the article, and approved the final draft.
- Shahd Al Ghawi performed the experiments, analyzed the data, prepared figures and/or tables, authored or reviewed drafts of the article, and approved the final draft.
- Mohammed Al-Badi performed the experiments, analyzed the data, prepared figures and/or tables, authored or reviewed drafts of the article, and approved the final draft.

### Human Ethics

The following information was supplied relating to ethical approvals (*i.e.*, approving body and any reference numbers):

The Medical Research Ethics Committee (MREC), College of Medicine and Health Sciences, Sultan Qaboos University, Oman, with an ethical approval number MREC #2920.

### Data Availability

The raw data is available at Zenodo: Alwahaibi, N. (2024). Raw data and other files [Data set]. Zenodo. https://doi.org/10.5281/zenodo.14160278.

## Supplemental Information

Supplemental information for this article can be found online at http://dx.doi.org/10.7717/peerj.20133#supplemental-information.

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
