# Peer review of "Knowledge and attitudes of university staff toward organ donation: a cross-sectional study in Oman"

_PeerJ, doi:10.7717/peerj.20133_

## Round 0.1 · original submission · Minor Revisions

·

Basic reporting

The manuscript is well-organized and generally written in clear and professional English. The introduction offers relevant context.

Experimental design

The study addresses a relevant public health issue – organ donation – using a well-structured cross-sectional design. The inclusion and exclusion criteria are appropriate, and the sampling method is described adequately.

Validity of the findings

The statistical methods (Chi-square, binary logistic regression) are appropriate for the research questions. The statistical methods (Chi-square, binary logistic regression) are appropriate for the research questions.

Additional comments

1. The description of the questionnaire could be improved.
2. The manuscript commendably outlines limitations, but should further discuss the potential impact of response/social desirability bias due to self-reporting on sensitive topics like organ donation.
3. The manuscript would benefit from a more detailed overview of the current organ donation system in Oman. Specifically, the authors should clarify whether Oman follows an opt-in or opt-out model. Discussing the regulatory framework and the operational status of national donor registries or transplantation networks would strengthen the relevance and interpretation of the findings.
4. In the Introduction, the authors refer to Spain, the USA, Portugal, Belgium, and Slovenia as “developing countries,” which is factually incorrect. These are high-income nations with mature healthcare systems. This phrasing should be corrected, and the text revised accordingly to reflect appropriate comparisons between developed and developing countries.
5. Did authors conduct any factor analysis for knowledge scale?
6. Table 1 is currently difficult to interpret due to its extensive categorical breakdowns. It is recommended that the table be restructured.
7. The line stating the “Total number of participants” at the bottom of tables is redundant. It would be more concise and professional to include the total sample size (e.g., N=385) in the table caption rather than as a separate line within the table.
8. The results section refers to significant associations using chi-square tests but does not report the actual test statistics (e.g., χ², df, p-values). These values should be provided in the tables or text to meet standard reporting practices.
9. For the binary logistic regression analyses, the manuscript currently presents adjusted odds ratios and confidence intervals, which is commendable. However, the authors should also report model-level statistics such as the model chi-square (or F statistic if linear regression is used), p-values, and R² to provide a fuller picture of model performance and explanatory power.
10. Tables 8 and 9 are informative, listing reasons for support or refusal of organ donation. To enrich the analysis, the authors could cross-tabulate these reasons with key demographic or professional variables. For instance, were religious motivations more common among certain job groups or age ranges? This would add depth to the discussion and help tailor future educational interventions more effectively.
11. The authors mention that the majority of respondents obtained information from online sources, but it is unclear whether the source of information was statistically associated with knowledge or attitude levels. Analyzing whether specific sources (e.g., healthcare professionals vs. social media) correlate with positive or negative attitudes would be valuable for designing future awareness strategies.
12. The Discussion would be more complete if the authors added a brief subsection on Implications. This should include recommendations for educational interventions, policy development, and integration of organ donation awareness into institutional programs, especially targeting administrative staff or male employees who demonstrated lower knowledge and attitude scores.
13. Suggested references:
1. Organ Donation Consent Systems: https://doi.org/10.2147/RMHP.S270234, https://doi.org/10.1136/bmjopen-2021-057107
2. Public and Professional Awareness and Attitudes: https://doi.org/10.3389/fpubh.2025.1551380, https://doi.org/10.21315/mjms2024.31.1.16, https://doi.org/10.1186/s12889-025-22044-4, https://doi.org/10.3389/fpubh.2025.1602008,
3. Religion in Muslim Communities: https://doi.org/10.1016/j.trre.2024.100874

·

Basic reporting

Reviewer comments: Dr Britzer Paul Vincent

Firstly, thank you for inviting me to review your paper titled “Knowledge and attitudes of university staff toward organ donation: a cross-sectional study in Oman”. This is an interesting piece of work, and I am delighted to see that the Deanship of Research at Sultan Qaboos University in Oman has commissioned it. The method adopted to conduct this study also shows the amount of work that had gone through to producing this piece of work. With regard to the religious legitimacy findings, it clearly resonates with our recently published systematic review, titled ‘Barriers and facilitators of deceased organ donation among Muslims living globally: An integrative systematic review.’
I have a few minor comments for the team to address before the acceptance for publication. They are as follows:

1. Line 88-92: Please give the reference for this sentence.
2. Line 92-94: You have mentioned “The highest organ donation rates per million population are observed in developing countries like Spain, the USA, Portugal, Belgium, and Slovenia.”. I think it is a typographical error as it should read as developed countries.
3. Line 102-104: Please give the reference for these two sentences.
4. Line 108-109: Please give the reference for this sentence.
5. Line 117-119: Could you give reference to this study please?
6. Line 132-133: Please give the reference for this sentence.
7. Line 158-169: The heading does not look relevant to the content below. The content describes more about the study setting which is the Sultan Qaboos University. And later the second half of the paragraph describes the data collection method. Since you have a separate section of data collection below, it is best to change the heading to study setting and have information relevant only to the study setting.
8. Line 315: Spelling error. Doners should be corrected to donors.
9. Line 32-323: You describe that the association between sex and knowledge is inconsistent across various studies. Any reason for this on why they findings are inconsistent?
10. In the discussion section it would be good to have a set of recommendation (either as text or a table) that you have particularly for Oman.

Thank you once again for your commendable work in the field of organ donation in Oman. I was pleased to read the updated version of your manuscript. For your interest, I have included a recently published paper below which may support or enrich the findings of your current study.

- Al-Abdulghani A, Vincent BP, Randhawa G, Cook E, Fadhil R. Barriers and facilitators of deceased organ donation among Muslims living globally: An integrative systematic review. Transplant Rev (Orlando). 2024;38(4):100874. doi:10.1016/j.trre.2024.100874

Experimental design

Methods used are relevant to the cross-sectional approach.

Validity of the findings

Relevant statistical tests used

In the discussion section it would be good to have a set of recommendation (either as text or a table) that you have particularly for Oman.

---

## Round 0.2 · Minor Revisions

I agree with the reviewer that there are so many tables in this manuscript. The authors should select important tables to use in the main text and put other tables in the appendix if needed.

·

Basic reporting

There are a lot of tables. It is recommended to move some of them to Suppl materials according to their importance.

Experimental design

-

Validity of the findings

-

---

## Round 0.3 · accepted · Accept

Thank you for addessing all the comments from the reviewers. The manuscript is ready for publication. Congratulations!